# Characterization and Quantitation of the Tumor Microenvironment of Uveal Melanoma

**DOI:** 10.3390/biology12050738

**Published:** 2023-05-19

**Authors:** Lara Goesmann, Nasrin Refaian, Jacobus J. Bosch, Ludwig M. Heindl

**Affiliations:** 1Department of Ophthalmology, Faculty of Medicine and University Hospital Cologne, University of Cologne, 50937 Cologne, Germany; lgoesman@smail.uni-koeln.de (L.G.); jbosch@chdr.nl (J.J.B.); ludwig.heindl@uk-koeln.de (L.M.H.); 2Center for Integrated Oncology (CIO) Aachen Bonn Cologne Duesseldorf, 50937 Cologne, Germany; 3Centre for Human Drug Research, 2333 CL Leiden, The Netherlands; 4Leiden University Medical Center, 2333 ZA Leiden, The Netherlands

**Keywords:** ocular melanoma, uveal melanoma, choroidal melanoma, tumor microenvironment, fluorescence immunohistochemistry, tumor-associated macrophages, immune checkpoint inhibitors

## Abstract

**Simple Summary:**

This is the first study on the tumor microenvironment of uveal melanoma with special regard to the localization within the tumor and the eye. Uveal melanoma is the most common malignant tumor of the eye in adults. Regardless of the treatment of the primary tumor, half of the patients with uveal melanoma die due to metastasis in the long run. The major findings of this study are that blood vessels are mainly located in the middle of the tumor and that immune cells (especially CD68-immunopositive macrophages) are mostly found in the outer section of the tumor. Furthermore, this study found a high representation of lymphocyte activation gene-3 (LAG-3) and Galectin-3 in uveal melanoma. These findings could set a foundation for new therapeutic strategies aimed at improving the survival rate of patients with uveal melanoma.

**Abstract:**

Uveal melanoma (UM) is a highly malignant tumor of the eye. Metastatic spread of UM occurs almost exclusively via blood vessels and is of tremendous interest, as half of the patients with uveal melanoma die of metastasis in the long run. The tumor microenvironment consists of all cellular and non-cellular compounds of a solid tumor, except for the tumor cells. This study aims to provide a more detailed understanding of the tumor microenvironment of UM to build the foundation for new therapeutic targets. Fluorescence immunohistochemistry was performed to examine the localization of various cell types in the tumor microenvironment in UM. Furthermore, the presence of LAG-3 and its ligands Galectine-3 and LSECtin was examined to evaluate the potential efficacy of immune checkpoint inhibitor-based therapies. The main findings are that blood vessels are mainly located in the middle of the tumor, and that immune cells are mostly found in the outer section of the tumor. LAG-3 and Galectine-3 were found to be highly represented, whereas LSECtin barely occurred in UM. Both the predominant location of tumor-associated macrophages in the outer section of the tumor and the high presence of LAG-3 and Galectine-3 in the UM serve as attainable therapeutic targets.

## 1. Introduction

Uveal melanoma (UM) is the most common and malignant tumor of the eye in adults. UMs arise from intraocular melanocytes in the uveal tract of the eye, which consists of the choroid, the ciliary body, and the iris. The majority of UMs are formed in the choroid (<90%), while the ciliary body and the iris are less commonly affected (>10%) [1,2,3]. The incidence rate of UM is between 2 and 8 cases per million people per year in Caucasians in Europe and the US, with most patients being older than 60 years [2,4,5,6]. In a recent study of the incidence and survival of UM in Germany, involving 1828 patients between 2013 and 2015, the age-standardized incidence was 5.1 per million people per year, with a median age of 67 years at diagnosis [3]. 

Various studies on the prognosis of UM have elaborated histopathological and cytogenic risk factors, such as UM of the epithelioid cell type, increased tumor-infiltrating macrophages, and tumor size. In addition, several other risk factors were elaborated, such as fair skin color, light iris color, and numerous skin naevi [1,7]. UMs harbor specific somatic chromosomal alterations and gene mutations. Approximately 90% of UMs have a driver mutation in either *GNAQ* or *GNA11*, which activates the MAPK signaling pathway. Other commonly occurring mutations found in UM are *BAP1* (39–51%), *EIF1AX* (12–21%), and *SF3B1* (10–22%) [7,8,9,10,11,12,13]. About 50% of UMs have monosomy 3, which is associated with the occurrence of metastasis and metastasis-related death, as up to 75% of patients with monosomy 3 develop metastases [14,15,16,17].

Metastatic spread of UMs occurs almost exclusively via blood vessels, with the liver being the most frequent metastatic site at about 90%. Other typical metastatic targets are the lungs at about 30% and the bones at about 15% [18,19]. Lymphatic metastasis is rare and occurs predominantly in cases of secondary extraocular tumor extension, which occurs in about 10% of UMs [20,21,22,23].

Current therapeutic options for primary UMs include bulbus-preserving therapies such as radiotherapy (brachy- or teletherapy), transpupillary thermotherapy, and surgical resection, as well as enucleation of the eye. Regardless of the treatment of the primary tumor, the prognosis is poor as half of the patients with UM die of metastasis in the long run [24,25,26]. The 10-year rates of metastasis of ciliary body melanoma were found to be 33%, whilst for choroidal melanoma it was 25% and 7% for iris melanoma [17]. However, none of the therapeutic options are effective when metastatic spread has taken place. This is largely owing to the lack of knowledge about how tumor cells interact with their respective tumor microenvironments [7].

The tumor microenvironment (TME) includes all cellular and noncellular compounds of a solid tumor, except for the tumor cells. It is a heterogeneous, complex, and versatile environment. There is constant modification of the TME as all components interact and influence the progression and spread of the tumor, whilst the tumor itself influences the TME simultaneously [27,28]. For example, inflammation caused by immigrating immune cells promotes angiogenesis, which eventually supports metastatic spread [29]. The heterogeneity of the TME has already been examined in various nonocular tumor entities, including breast cancer, lung cancer, and skin melanoma [30,31,32]. However, a more detailed understanding is required of how the TME of UM composes as well as how it interacts. 

The development of immune checkpoint inhibitor-based therapies has been a promising breakthrough in cancer treatment, for example, in breast cancer and malignant melanoma of the skin. Some immune checkpoint inhibitors have already been approved for certain cancer types, whilst others are currently undergoing clinical trials. Many studies have proven that tumor cells can modulate the immune response to escape antitumor activity by activating immune checkpoint pathways that induce immunosuppressive functions [33,34]. To date, UMs have been very unresponsive to checkpoint immunotherapy [35,36,37]. Thus, a deeper knowledge of checkpoints in UMs is needed to make UMs amenable to immune checkpoint inhibitor-based therapies.

Lymphocyte activation gene-3 (LAG-3) is a type I transmembrane protein expressed on the surface of effector T-cells and regulatory T-cells. LAG-3 serves as an inhibitory coreceptor that co-localizes with CD4, CD8, and CD3 molecules to prevent excessive immune responses [38,39]. In recent years, various studies have shown that the inhibition of LAG-3 allows T-cells to regain their cytotoxic function and that the suppressed immune response can be reduced, resulting in a regained antitumor effect [40,41]. Furthermore, LAG-3 has been identified as an indicator of tumor prognosis. In non-small-cell lung cancer, for instance, a high level of LAG-3 indicates a poor prognosis, whereas in gastric carcinoma and melanoma, it is associated with a better prognosis [42]. In UM, a high level of LAG-3 was found to be associated with high-risk tumor parameters [43].

Galectin-3 is a ligand of LAG-3 that modulates the T-cell response via several mechanisms. In vitro experiments have shown that Galectin-3 needs to bind LAG-3 for Galectin-3-mediated suppression of CD8-positive T-cells. Therefore, inhibition of Galectin-3 through an immune checkpoint inhibitor should result in T-cell activation and a stronger tumor-specific immune response [39,44].

LSECtin is a type II transmembrane protein that has been detected in various human tumors and is a ligand of LAG-3 as well. Increased expression of LSECtin has been found in bladder, gastroesophageal, and melanoma tumors. The LSECtin/LAG-3 interaction inhibits IFN-γ secretion from T-cells, resulting in a reduced antitumor immune response [45]. Thus, inhibition of the LSECtin/LAG-3 complex through immune checkpoint inhibitors could help increase the antitumor response by restoring the IFN-γ secretion from effector T-cells. 

## 2. Materials and Methods

### 2.1. Tissue Sample Selection and Processing

The study cohort included 18 patients with choroidal UM who underwent enucleation in the Department of Ophthalmology at the University of Cologne, Germany between 2007 and 2017. With approval for scientific examination from the local Ethics Committee of the University of Cologne (Medizinische Fakultaet der Universitaet zu Koeln—Geschaeftsstelle der Ethikkommission; 15–362; 26 July 2016), the collection was obtained from the laboratory for histology and ophthalmopathology. An expert pathologist reviewed all cases and confirmed the diagnosis. All tumor characteristics were obtained from the pathology reports. 

The enucleated eyes were formalin-fixed and paraffin-embedded. Afterwards, multiple four μm-thick serial sections were cut from the tissue blocks for subsequent analyses.

### 2.2. Fluorescence Immunohistochemistry (IHC)

In all cases, we performed IHC using a double-staining technique under optimized conditions, including specified antigen retrieval, autofluorescence quenching, concentrations, incubation times, and incubation temperatures. Melanoma Pan, a ready to use antibody mix consisting of anti-HMB45-, anti-Mart-1/Melan A-, and anti-Tyrosinase-antibody was applied on all slides to prove melanoma cells and define the exact tumor area. Additionally, a second primary antibody was applied to prove the presence of another specific cell type represented in the TME.

Staining protocol: Firstly, deparaffinization was performed using xylene. The slides were rehydrated using 100%, 70%, and 50% ethanol and washed with distilled water afterwards. Antigen retrieval was performed with either Fast Enzyme (CD68 and Pan-CK) or with Retrieval Solution (CD31, CD3, LYVE1) for 30 min at 96 °C. For autofluorescence quenching, True Black (1:20 in 70% ethanol) was applied to all slides for 15 seconds. 5–10% normal serum (host species of the secondary antibody) diluted in PBS was applied to all slides for 10 min. The primary antibodies were diluted in antibody dilution buffer, and the mixture was applied to the slides according to the concentrations, incubation times, and temperatures listed in Table 1. The secondary antibodies (Invitrogen, goat anti-mouse IgG Alexa Fluor 488 and Invitrogen, goat anti-rabbit IgG Alexa Fluor 546) were diluted in PBS (1:250) and the samples were incubated for 60 min in the dark. Nucleus staining was performed with DAPI diluted in PBS (1:2000), and the mixture was applied to the slides for 2 min. Fluorescence Mounting Medium was applied, and the slides were covered with coverslips and stored at approximately 4 °C in the dark.

Every incubation step was performed at room temperature unless stated otherwise. After every incubation step, the slides were washed three times with PBS for 5 min each.

### 2.3. Image Acquisition and Analysis

The stained slides were imaged using a Zeiss confocal microscope and Zeiss Zen Blue software at 25× magnification. The areas of image acquisition were determined beforehand based on the localization within the tumor (outer section, intermediary section, and inner section) and on the localization within the eye (sclera side, middle part, and vitreous body side) (Figure 1). Accordingly, a total of 14 images per slide were taken.

Image segmentation was carried out using the machine-learning segmentation tool Ilastik with the respective workflow being Pixel Classification. By applying this workflow, we assigned labels to pixels and trained the classifier to separate the object classes. This ensured objective and reproducible results. The area covered by immunopositive cells was determined and analyzed.

### 2.4. Statistical Analysis

All statistical analyses were performed using the GraphPad Prism in consultation with a medical statistician at the University of Cologne. For each tumor, the results were categorized as paired or nonparametric. Thus, the Wilcoxon test or Friedman test was performed throughout. 

## 3. Results

The tumor characteristics at enucleation are listed in Table 2. The patients’ median age was 69 and gender was equally distributed. The cohort consisted of enucleated eyes with choroidal UM, which had a mean basal diameter of 17.21 mm and a mean height of 7.37 mm. Immunohistochemistry was performed after enucleation. The spindle cell type was found in 47% of cases, the epithelioid cell type was found in 13% of cases, and the mixed cell type was found in 40% of cases. A semiquantitative classification of the melanization of the tumor from + to ++++ showed that 44.44% of the tumors were strongly pigmented (+++ and ++++), whereas 55.55% of the tumors were less pigmented (+ and ++). 

Figure 2 shows the double fluorescence immunohistochemistry results for the specific markers. The distribution of the various cell types and their percentage of covered area within the tumor are depicted in Figure 3. We found major differences in the distribution of the cell types, with melanoma cells and immune cells being the predominant ones. Among the immune cells, macrophages were more abundant than T-lymphocytes. Within the tumor, an average of 4.93% was covered by CD68-immunopositive macrophages and 1.33% by CD3-immunopositive T-lymphocytes. The differences between these cell types were found to be nonsignificant (*p* = 0.750).

CD31-immunopositive blood endothelial cells covered a mean area of 0.39% and were significantly less present than CD68-immunopositive macrophages (*p* = 0.008). However, LYVE1-immunopositive lymph endothelial cells and CD31-immunopositive epithelial cells were significantly less abundant than immune cells, they covered a mean area of 0.06% each (LYVE1 vs. CD68, *p* < 0.001; LYVE1 vs. CD3, *p* = 0.002; CD31 vs. CD68, *p* = 0.008; CD31 vs. CD3, *p* > 0.9999).

In order to clarify whether the distribution of the cell types is influenced by their localization within the tumor, we categorized the tumor into outer, intermediary, and inner sections. The results are illustrated in Figure 4. 

Most of the cell types showed significant difference between the outer section and the inner sections; immune cells and lymph endothelial cells were mostly located in the outer section, whereas blood endothelial cells were predominantly present in the inner section. Compared to the inner section of the tumor, a significantly larger area was allocated to CD68-immunopositive macrophages (*p* = 0.003) and CD3-immunopositive T-lymphocytes (*p* = 0.014) in the outer section. Additionally, a significantly larger area was covered by LYVE1-immunopositive cells in the outer section compared to the inner section (*p* = 0.003). Furthermore, none of the cell types showed significant differences between the inner section and the intermediary section. Epithelial cells showed no significant differences in distribution.

Apart from the localization within the tumor, we also paid attention to the distribution of the cell types based on the localization within the eye. We did this by subdividing the tumor into a sclera side, a middle part, and a vitreous body side (Figure 5).

The immune cell types showed a similar distribution pattern. Compared to the middle of the tumor, more immune cells were localized in the vitreous body side and the sclera side. However, significantly more CD68-immunopositive macrophages were detected in the sclera side compared to the middle section (*p* = 0.037), whereas significantly more CD3-immunopositive T-lymphocytes were detected in the vitreous side compared to the middle section (*p* = 0.005). Interestingly, CD31-immunopositive blood endothelial cells covered significantly more area in the middle section compared to the sclera (*p* = 0.005) and vitreous body side (*p* = 0.003). LYVE1-immunopositive lymph endothelial cells were mostly located in the vitreous body side, with a significant difference compared to the middle section (*p* = 0.008). None of the cell types presented a significant difference in distribution between the sclera side and the vitreous body side of the tumor. In addition, no significant difference in distribution was found for epithelial cells.

Figure 6 shows the fluorescence immunohistochemistry of LAG-3, LSECtin, and Galectin-3. The overall results for the area covered by the stated markers are depicted in Figure 7. Galectin-3 occurred the most, with an average of 2.48% of the area being covered, followed by LAG-3 with 0.87% of the covered area. The difference between these cell types was found to be nonsignificant (*p* > 0.9999). Almost no area was covered by LSECtin (0.002%), and the differences between LSECtin and the other two markers were found to be significant (LAG-3 *p* = 0.0003, Galectin-3 *p* < 0.0001).

Our previous examination has shown significant differences in the distribution of various cell types between the outer and inner sections. Accordingly, we subdivided the tumors into these sections to investigate the distribution of LAG-3, LSECtin, and Galectin-3. The results are summarized in Figure 8, showing no significant differences in the distribution of the previously mentioned. (LAG-3: outer section vs. inner section, *p* = 0.3289. LSECtin: outer section vs. inner section, *p* = 0.6788. Galectin-3: outer section vs. inner section, *p* = 0.1297).

Furthermore, the distribution of LAG-3, LSECtin, and Galectin-3 based on their localization within the eye has shown no significant differences. (LAG-3: sclera side vs. middle, *p* > 0.9999; middle vs. vitreous body side, *p* = 0.6898; sclera side vs. vitreous body side, *p* > 0.9999. LSECtin: sclera side vs. middle, *p* > 0.9999; middle vs. vitreous body side, *p* => 0.9999; sclera side vs. vitreous body side, *p* > 0.9999. Galectin-3: sclera side vs. middle, *p* = 0.4719; middle vs. vitreous body side, *p* > 0.9999; sclera side vs. vitreous body side, *p* > 0.9999). The results are depicted in Figure 9.

## 4. Discussion

The present study is the first investigation of the composition of the TME of choroidal melanoma with respect to the localization within the tumor as well as within the eye. The aim is to set the foundation for new therapeutic strategies and targets in order to eventually improve the survival rate of patients with UMs.

We discovered the following new and important results: blood vessels are mainly located in the middle of the tumor, whereas LYVE1-immunopositive cells (lymph endothelial cells) and immune cells are mostly found in the outer section of the tumor. Within the outer section, CD68-immunopositive macrophages, CD3-immunopositive T-lymphocytes, and LYVE1-immunopositive cells occupied more area in the vitreous body side compared to the sclera side.

In our study cohort of 18 enucleated eyes, the median age of the patients at enucleation was 69 years. Recently, the majority of UMs were reported to occur at above 65 years of age in Germany, which coincides with our results [3]. Male and female patients were equally represented in our study group, supporting the results of a large study of 8033 eyes with UMs by Shields et al [46]. 

As the mean basal diameter of our study cohort was 17.21 mm, with a mean tumor height of 7.37 mm, all conclusions made from our results are solely applicable to large-sized tumors. The estimated 5-year mortality rates of patients with UM were found to be much higher (53%) for patients with large tumors (basal diameter > 15 mm and tumor thickness > 8 mm), than for patients with smaller tumors (16% for small tumors and 32% for medium-sized tumors) [1,47]. Furthermore, Shields et al. showed that each millimeter increase in tumor thickness was associated with an approximately 5% increase in the risk for metastasis after 10 years [46].

In our study cohort, the spindle cell type was found in 47% of cases, the epithelioid cell type in 13%, and the mixed cell type in 40% of cases. Tumor cell type is an important prognostic factor for UM. The Callender classification system subdivides UM into spindle A, spindle B, epithelioid, and mixed tumors [48]. Various studies have proven that UM of the spindle cell type has the best prognosis, whereas epithelioid cell melanoma has the worst prognosis [49,50,51,52]. We found that tumors of the epithelioid cell type were of a larger mean size (20.5 mm) than those of the spindle cell (15.66 mm) or mixed cell type (17.66 mm). We hypothesized that there is a positive correlation between the quantity of epithelioid cells and tumor growth, resulting in larger tumor sizes at enucleation of UM of the epithelioid or mixed cell type.

We found that the mean tumor area covered by CD68-immunopositive macrophages was 4.94%. Interestingly, the mean area in epithelioid cell tumors was 8.97%, in the mixed type it was 4.41%, and 3.04% in the spindle cell type. Makitie et al. found that a high quantity of tumor-infiltrating CD68-immunopositive macrophages is associated with the presence of epithelioid cells [53]. Our results support these findings, and we conclude that the quantity of epithelioid cells in a tumor correlates strongly with the number of CD68-immunopositive macrophages, and therefore with tumor malignancy.

Macrophages have been investigated in various malignant cancer entities, such as ovarian tumors, pancreatic, rectal, and breast cancers [54,55,56,57]. Depending on the type of macrophages, they either provide defense against tumor cells (M1-type) or promote tumor cell proliferation (M2-type). Tumor-associated macrophages have been proven to mainly exhibit the M2 phenotype and are associated with the promotion of tumor cell proliferation, metastasis, and angiogenesis [58,59,60,61,62]. In this study, we found that CD68-immunopositive macrophages make up a large portion of the TME. Bronkehorst et al. have shown that most macrophages in UM are of the M2-type [61]. Based on these data and our finding that macrophages are the most abundant cell type after melanoma cells, the severe malignancy of UM may be due to the high presence of M2-macrophages. Hence, focusing on macrophages as therapeutic targets may play an important role in the treatment of UM in the future.

We also found that macrophages were mainly located in the outer section of the tumor. As M2-macrophages are proven to be related to tumor growth, it might be of interest to deplete or even erase these cells. Bisphosphonates (e.g., liposome-encapsulated clodronate) have shown great efficacy in macrophage depletion in in vitro and in in vivo mouse models [63,64,65]. For the purpose of macrophage depletion, high doses of systemic therapy would be necessary because the in vivo efficacy is limited by a short plasma half-life or by being excreted unaltered by the kidneys. However, such high doses cannot be applied in humans because of nephrotoxicity, osteonecrosis, and bone fractures [66,67]. Due to the predominant localization of macrophages in the outer section of choroidal melanoma, we conclude that it may be an efficient therapeutic option to undertake an intravitreal injection with a bisphosphonate of lower dose to achieve a sufficient local dose for macrophage depletion. This may stop tumor growth and lead to downsizing of the tumor, making it more responsive to thermo-, photodynamic-, or radiotherapy. In a study of 20 eyes of 20 pigmented rats, Nourinia et al. showed that up to 8 μg of intravitreal zoledronic acid seems to be safe in rat eyes [68]. The next steps for investigating this new therapeutic approach would be to examine the most suitable bisphosphonate, the necessary dose for macrophage depletion, and the maximum applicable in vivo dose in human eyes. We do not consider a bisphosphonate injection into the tumor itself, as this could cause cell seeding within the eye [69]. 

Another major finding of our study was the predominant localization of blood endothelial cells in the middle part of the tumor. Generally, in the early stages of tumor formation, hyperplastic growth takes place. Having reached a critical size, pre-existing blood vessels cannot provide sufficient oxygen and nutrients. Thereby, further tumor growth is impaired. However, tumor cells can overcome this stagnation by inducing tumor angiogenesis. The transition from initial hyperplasia to vascularized malignant tumor growth has been described and investigated as an “angiogenic switch” by several authors [70,71,72]. As a matter of fact, the formation of blood vessels plays a crucial role in the progression and differentiation of the TME. Having found that blood vessels in UMs are mainly located in the center of the tumor, we conclude that the formation and differentiation of the TME takes place radially, which ensures sufficient oxygen and nutrient supply in all parts of the tumor. Based on this, the tumor center must be the longest existing part of the tumor with the most time for angiogenesis, resulting in a high microvascular density. The major localization of blood vessels in the tumor center makes it difficult to treat choroidal melanoma from the outside. For choroidal melanomas of small sizes, this is carried out via transpupillary thermotherapy, photodynamic therapy, plaque radiotherapy, or proton beam radiotherapy. However, large tumor size limits the success of these therapies, and we conclude that this is due to the central localization of the blood vessels.

Besides epithelial cells, lymph endothelial cells occurred the least out of all cell types in our study. Furthermore, LYVE1-immunopositive cells were mostly found in the outer section of the tumor. LYVE1is not only expressed in lymphatic endothelial cells but also in macrophages and various structures in the anterior segment of the eye [23,73]. Moreover, Schroedl et al. showed in their study of 17 normal eyes that the adult human choroid does not contain typical lymphatic vessels but a significant amount of LYVE1immunopositive macrophages [73]. Based on the small amount of LYVE1-immunopositive cells in our findings and the same predominant localization as determined for CD68-immunopositive macrophages, the functionality of these structures as lymphatic vessels remains vague. In the event that these structures were not lymphatic vessels, this would support various studies that could not detect intraocular lymphatic structures in UM [19,23,74]. 

The examination of LAG-3 and its ligands showed a high representation of Galectin-3 and LAG-3, whereas LSECtin was barely represented. Kashyap et al. that found high levels of LAG-3 were associated with UM with high-risk parameters and high metastatic potential. The large tumor sizes in our study group combined with the high representation of LAG-3 support this finding [75]. Furthermore, our results support the assumption made by Souri et al. that LAG-3 and Galectin-3 interact strongly in UM. Galectin-3 is known to bind LAG-3 on CD8-positive T-cells. Thus, the LAG-3/Galectin-3 interaction in UM may result in depleted activity of CD8-positive T-cells [43]. Durante et al. found LAG-3 to be the most dominant immune checkpoint in UM [76]. In our examination, the difference in the quantity of LAG-3 and Galectin-3 was found to be insignificant.

Immune checkpoint inhibitor-based therapies are a modern and promising approach for the treatment of various solid tumors, including metastatic cutaneous melanoma. In contrast, UM has shown a poor response to checkpoint immunotherapies such as cytotoxic T-lymphocyte antigen 4 (CTLA-4) and programmed cell death protein 1 (PD-1) [33,34,35,37,77,78]. However, high expression of PD-1 and programmed cell death-ligand 1 (PD-L1) and the presence of tumor-infiltrating lymphocytes are associated with parameters for poor prognosis, such as loss of BAP-1, the epithelioid cell type, and liver metastasis [79]. Similarly, methylation of immune checkpoint genes, such as CTLA4, PD-1, PD-L1, and LAG-3, has been shown to correlate strongly with BAP1 mutation status and overall survival in UM. Therefore, DNA methylation tests for immune checkpoint genes should be used as predictive biomarkers for response to immunotherapy [80].

Considering the mean area covered by the various cell types, we analyzed an average of 13% of the total tumor area. Obviously, the TME consists of additional cell types and subtypes as well as noncellular compounds, which could not all be covered by the marker panel applied in this study. Moreover, one needs to be aware that the presented results arise from two-dimensional tumor slices, whereas the tumor itself is a complex three-dimensional construct. In summary, we have shown remarkable differences in the localization and the occurrence of various cell types in the TME of uveal melanoma. Based on these findings, a larger study group, an extended marker panel, and preclinical studies are required to support our results and explore new treatment approaches. 

## 5. Conclusions

In the present study we have proven that the distribution of different cell types of the TME of UM shows remarkable differences in localization; blood vessels are mainly located in the middle of the tumor, whereas immune cells (especially CD68-immunopositive macrophages) are mostly found in the outer section of the tumor. LAG-3 and Galectin-3 have been equally represented in our findings, indicating that the inhibition of Galectin-3 through an immune checkpoint inhibitor may be a reasonable approach in UM treatment. These findings may pave the way for new therapeutic strategies aiming to improve the survival rate of patients with uveal melanoma.

## Figures and Tables

**Figure 1 biology-12-00738-f001:**
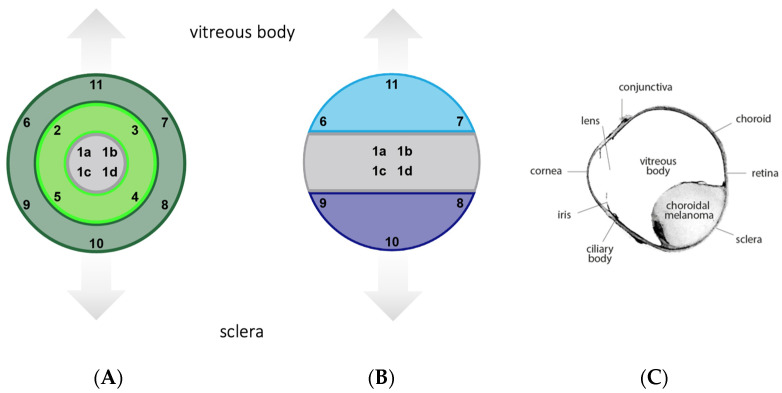
Template for image acquisition. (**A**) Based on the localization within the tumor. Inner section (images 1a–d), intermediary section (images 2–5), outer section (images 6–11) (**B**) Based on the localization within the eye. Middle part (images 1a–d), sclera side (images 8–10), vitreous body side (images 6, 7, 11). (**C**) Overview of a histological specimen.

**Figure 2 biology-12-00738-f002:**
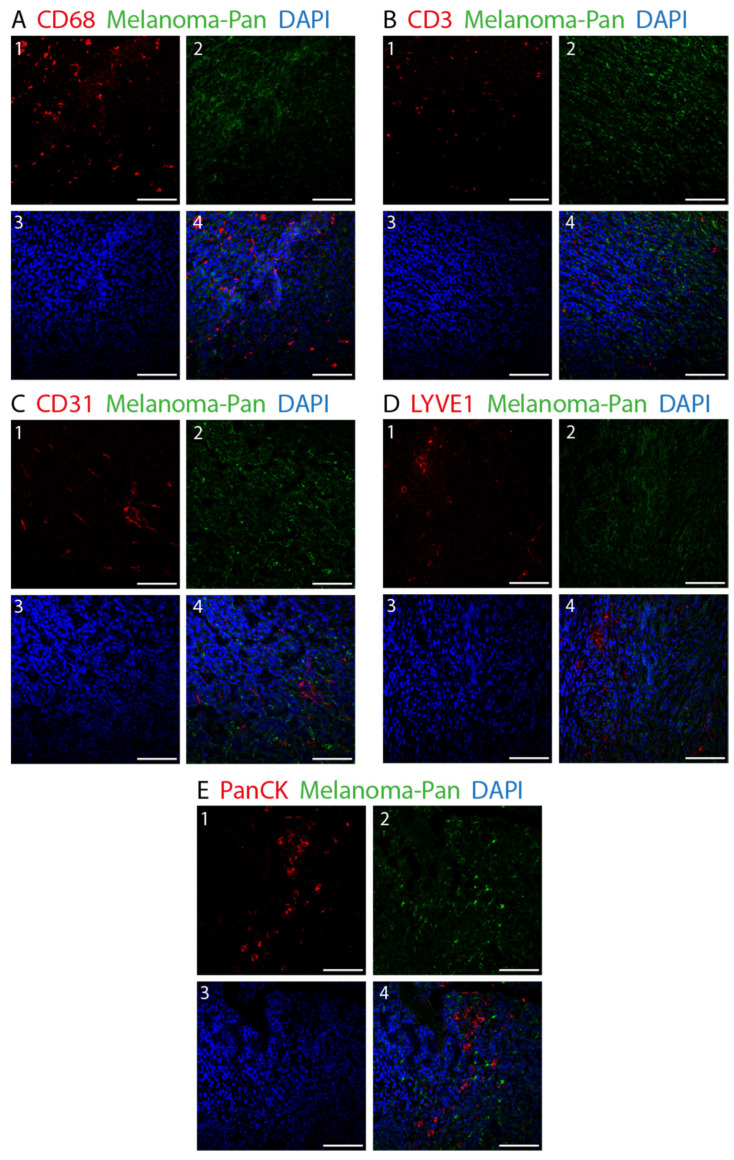
Double fluorescence IHC staining in choroidal melanoma. Melanoma Pan (green) was applied on all slides to define the exact tumor area. Various other cell types were identified using different markers (red). (**A**) 1: CD68-immunopositive macrophages. (**B**) 1: CD3-immunopositive T-lymphocytes. (**C**) 1: CD31-immunopositive blood endothelial cells. (**D**) 1: LYVE1-immunopositive lymph endothelial cells. (**E**) 1: PanCK-immunopositive epithelial cells. (**A**–**E**) 2: Melanoma Pan-immunopositive melanoma cells. (**A**–**E**) 3: Dapi. (**A**–**E**) 4: Merged image. (×25, scale bar = 100 μm).

**Figure 3 biology-12-00738-f003:**
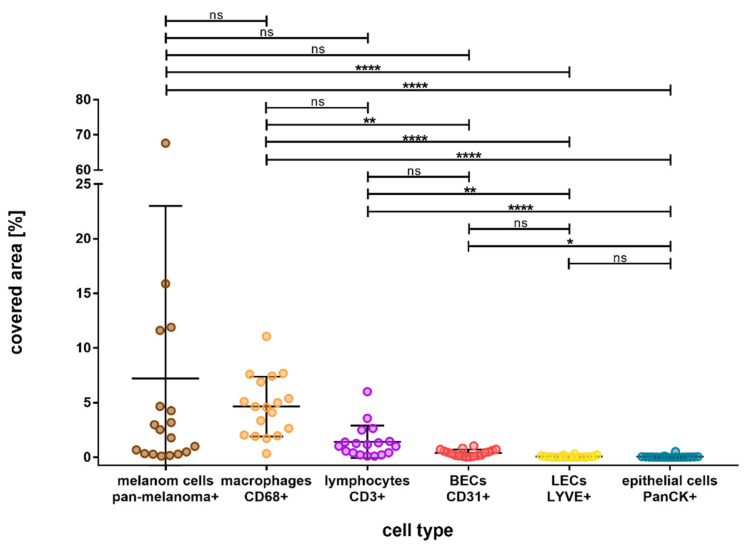
Overall outcome of the percentage area covered by different cell types in choroidal melanoma, arranged in the order from most to least covered area. Melanoma Pan-immunopositive melanoma cells and CD68- and CD3-immunopositive immune cells were represented the most, followed by CD31-immunopositive blood endothelial cells (BEC). LYVE1-immunopositive lymph endothelial cells (LEC) and PanCK-immunopositive epithelial cells were the least represented cell types. Significance levels: ns = *p* > 0.05, * = *p* ≤ 0.05, ** = *p* ≤ 0.01, and **** = *p* ≤ 0.0001.

**Figure 4 biology-12-00738-f004:**
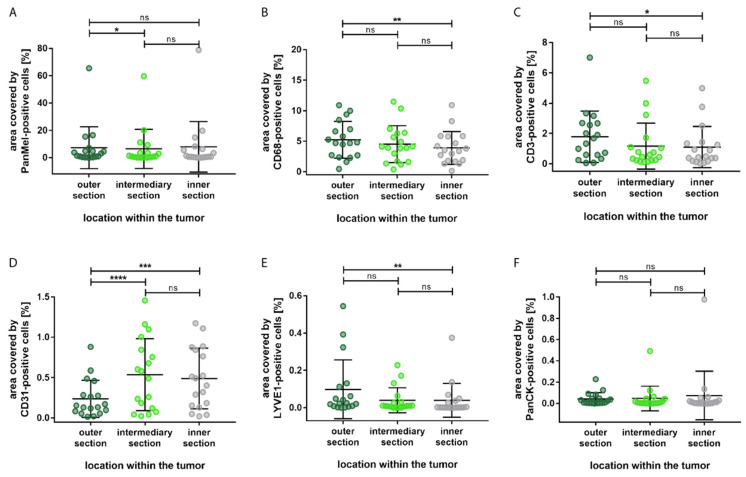
Distribution of the percentage area covered by different cell types based on the localization within the tumor. (**A**) Significantly more Melanoma Pan-immunopositive melanoma cells in the outer section compared to the intermediary section. (**B**) Significantly more CD68-immunopositive macrophages in the outer section compared to the inner section. (**C**) Significantly more CD3-immunopositive T-lymphocytes in the outer section compared to the inner section. (**D**) Significantly more CD31-immunopositive blood endothelial cells in the inner section and in the intermediary section compared to the outer section. (**E**) Significantly more LYVE1-immunopositive lymph endothelial cells in the outer section compared to the inner section. (**F**) No significant differences in the distribution of PanCK-immunopositive epithelial cells. Significance levels: ns = *p* > 0.05, * = *p* ≤ 0.05, ** = *p* ≤ 0.01, *** = *p* ≤ 0.001 and **** = *p* ≤ 0.0001.

**Figure 5 biology-12-00738-f005:**
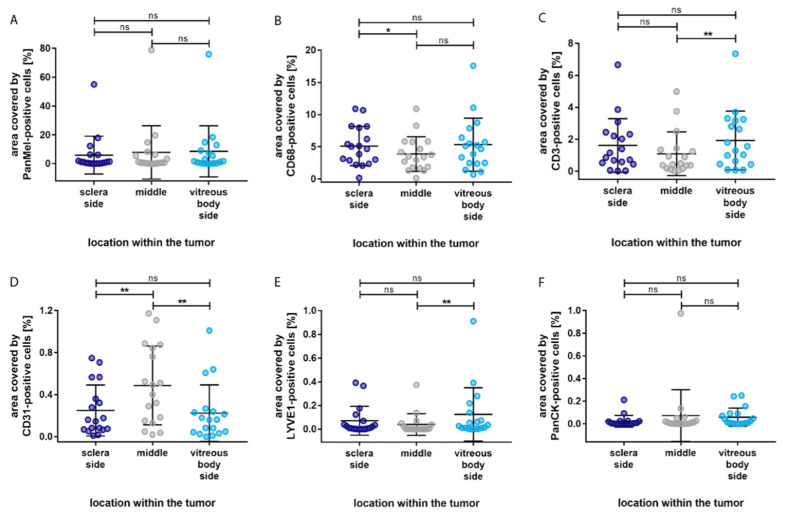
Distribution of the percentage area covered by different cell types based on the localization within the eye. (**A**) No significant difference in the distribution of Melanoma Pan-immunopositive melanoma cells. (**B**) Significantly more CD68-immunopositive macrophages in the sclera side compared to the middle section. (**C**) Significantly more CD3-immunopositive T-lymphocytes in the vitreous body side compared to the middle section. (**D**) Significantly more CD31-immunopositive blood endothelial cells in the sclera side and in the vitreous body side compared to the middle section. (**E**) Significantly more LYVE1-immunopositive lymph endothelial cells in the vitreous body side compared to the middle section. (**F**) No significant differences in the distribution of PanCK-immunopositive epithelial cells. Significance levels: ns = *p* > 0.05, * = *p* ≤ 0.05, and ** = *p* ≤ 0.01.

**Figure 6 biology-12-00738-f006:**
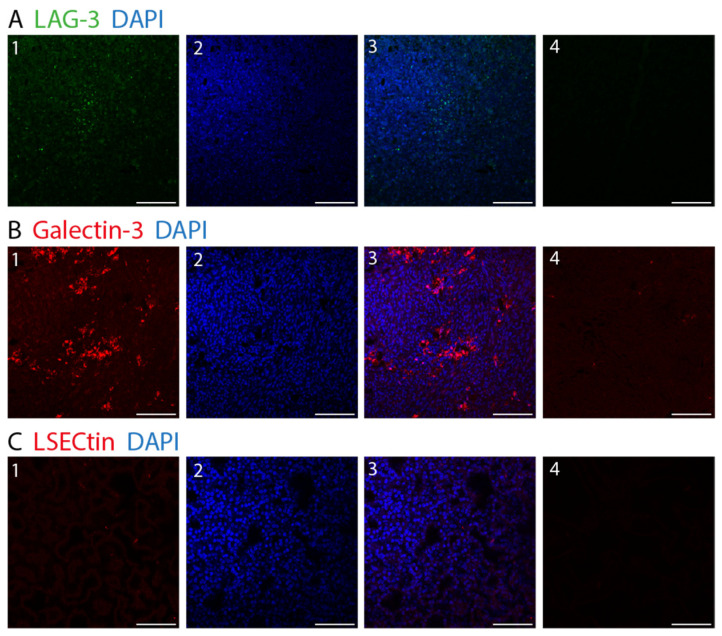
Fluorescence IHC staining in choroidal melanoma. (**A**) Lymphocyte activation gene-3. (LAG-3) (1: LAG-3 (green), 2: DAPI (blue), 3: LAG-3 (green) and DAPI (blue)). (**B**) Galectin-3 (1: Galectin-3 (red), 2: DAPI (blue), 3: Galectin-3 (red) and DAPI (blue)). (**C**) LSECtin (1: LSECtin (red), 2: DAPI (blue), 3: LSECtin (red) and DAPI (blue)) (**A**–**C**) 4: negative control. (×25, scale bar = 100 μm).

**Figure 7 biology-12-00738-f007:**
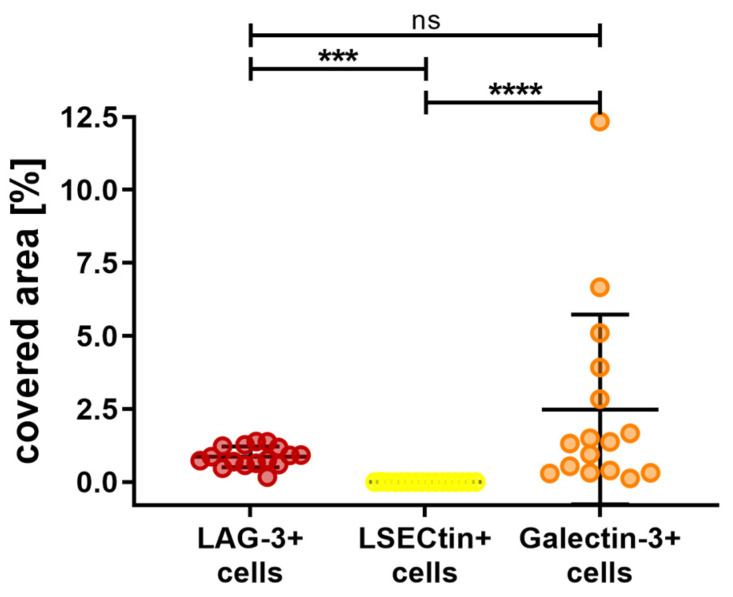
Overall outcome of the percentage area covered by LAG-3, LSECtin, and Galctine-3 in choroidal melanoma. Galectine-3-positive cells were represented the most, followed by LAG-3-positive cells. LSECtin-positive cells were the least abundant. Significance levels: ns = *p* > 0.05, *** = *p* ≤ 0.001 and **** = *p* ≤ 0.0001.

**Figure 8 biology-12-00738-f008:**
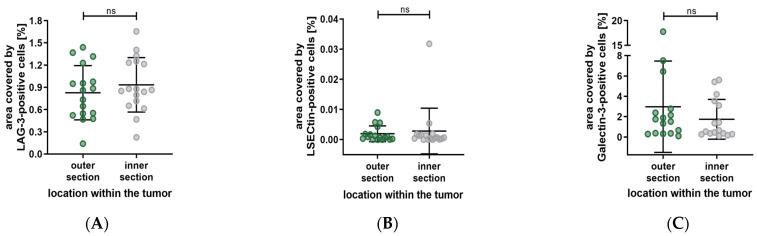
Distribution of the percentage area covered by LAG-3, LSECtin and Galectin-3 based on the localization within the tumor. (**A**–**C**) No significant difference in the distribution of LAG-3, LSECtin and Galectin-3. Significance level: ns = *p* > 0.05.

**Figure 9 biology-12-00738-f009:**
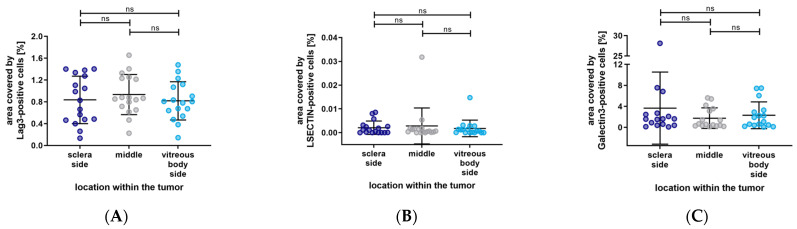
Distribution of the percentage area covered by LAG-3, LSECtin, and Galectin-3 based on the localization within the eye. (**A**–**C**) No significant differences in the distribution of LAG-3, LSECtin, and Galectin-3. Significance level: ns = *p* > 0.05.

**Table 1 biology-12-00738-t001:** List of primary antibodies, including concentrations, incubation times, temperatures, and identification numbers.

Primary Antibody	Concentration	Incubation Time	Temperature	Manufacturer Identification
Melanoma Pan	ready to use	30 min	RT	#DCS, MI875R06
CD31	1:50	30 min	RT	#abcam, ab76533
Pan Cytokeratin (Pan-CK)	1:100	30 min	RT	#origene, DP010
CD68	1:100	45 min	RT	#abcam, ab213363
CD3	1:75	45 min	RT	#DCS, CI597C002
LYVE1	1:25	over night	4 °C	#abcam, ab36993
LAG-3	1:100	60 min	RT	#novusbio, NBP1-97657
LSECtin	1:100	60 min	RT	#ThermoFisher, PA5-53116
Galectin-3	1:200	60 min	RT	#ThermoFisher, 14-5301-82

**Table 2 biology-12-00738-t002:** Tumor characteristics at enucleation of the eye. Semiquantitative classification of the tumor melanization from + (less pigmented) to ++++ (strongly pigmented).

Nr.	Age at Enucleation (n = 18)	Gender (n = 18)	Basal dia-Meter of Tumor (n = 14)	Hight of Tumor (n = 14)	Type of Tumor (n = 15)	Semiquantitative Classification of Melanization
1	44	f	15 mm	6.5 mm	spindle cell type	+++
2	80	m	22 mm	6.5 mm	spindle cell type	+++
3	69	f	15 mm	10 mm	epithelioid cell type	+
4	69	m	12 mm	4.3 mm	spindle cell type	+
5	79	f	26 mm	7.5 mm	epithelioid cell type	++
6	47	m	18 mm	7.5 mm	spindle cell type	+
7	65	f				+++
8	74	m	20 mm	12 mm	mixed cell type	+
9	83	m	13 mm	5 mm	spindle cell type	++
10	76	f	21 mm	5 mm	mixed cell type	++
11	62	f	15 mm	6 mm	mixed cell type	++
12	22	f	14 mm	11 mm	spindle cell type	+
13	89	m	13 mm	10.5 mm	mixed cell type	+++
14	47	m			spindle cell type	++
15	47	m	16 mm	6 mm		+++
16	86	f			mixed cell type	+++
17	47	f				+++
18	83	m	21 mm	5.4 mm	mixed cell type	++++

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
