# Peer review of "Characterization and Quantitation of the Tumor Microenvironment of Uveal Melanoma"

_biology, 2023, doi:10.3390/biology12050738_

Round 1
Reviewer 1 Report
Dear Editor,
Thank you for giving me the opportunity to review this paper. The authors have used a multiplexed IHC panel to characterise the tumour microenvironment of uveal melanoma tissue sections. It appears that the authors are not familiar with the concept of 'antibody validation'. Using commercial antibodies without validation data (except clinical grade antibodies) are unacceptable for publication. Also, quantification of images are not adequately descibed.
The findings are interesting. If the authors can show that the commercial antibodies are specific (antibody specificity) for these targets, this paper will then be presenting interesting and publication worthy data.
Author Response
Dear Reviewer 1,
Thank you for reviewing our manuscript and for your comments. Please see the attached cover letter (with attachments) with the point-by-point response to your comments.
Best regards,
Nasrin Refaian

Reviewer 2 Report
The article written by Lara Goesmann et al summerizes the distribution of different cell types of the tumor microenvironment (TME) of uveal melanoma with respect to the localization within the tumor. Blood vessels are mainly located in the middle of the tumor, and immune cells are mostly found in the outer section of the tumor. Futhermore, LAG-3 and Galactine-3 were found to be highly represented.
The topic of the article is very actual and interesting. The inhibition of Galactine-3 through an immune checkpoint inhibitor might function a reasonable approach in UM treatment. High percentage of LAG-3 and Galactine-3 in human UM can represent novel potential therapeutic strategies.
Generally, the design and structure of the work are appropriate. The figures are simply constructed and depicting the essential informations about the topic, sufficiently supporting the message of the text.
The results are more or less clearly presented. However, there are some major shortcomings and confusions, which would all require clear corrections.
In the MS Table 2. appears before Table 1. The first figure in the MS is Figure 6. then Figure 3,4 and 5, then another Figure 6. Later Figure 7 and 8. In addition, there is no Figure 2.
How can it be??
Author Response
Dear Reviewer 2,
Thank you for reviewing our manuscript and for your comments. Please see the attached cover letter (with attachments) with the point-by-point response to your comments.
Best regards,

Reviewer 3 Report
The manuscript entitled 'Characterization and quantitation of the tumor microenvironment of 3 uveal melanoma' is well written and executed. Experiments are well defined and performed. There are minor suggestion:
- Please add some latest references (PMID: 36918273, PMID: 35862127, PMID: 33136179) in discussion section.
- Please do some grammar check.
Author Response
Dear Reviewer 3,
Thank you for reviewing our manuscript and for your comments. Please see the attached cover letter (with attachments) with the point-by-point response to your comments.
Best regards,

Round 2
Reviewer 1 Report
Dear Author,
Clinical grade antibodies can be safely exempted from the validation process. But for other antibodies, forwarding the vendor's antibody data sheet is not the way to go forward. If you can replicate what vendors show on their data sheet is key. For that, one needs to test antibodies by western blot to see other non target bindings, IHC done on positive and negative control (cell lines naturally not expressing or overexpressing target gene, knockout/knocked down cell lines, overexpression cell lines, tissue control to make sure antibody specificity).
Thank you.
Reviewer 2 Report
After the revision and corrections the manuscript seems to be fine.